

# Hidden symmetry of Bogoliubov de Gennes quasi-particle eigenstates and universal relations in flat band superconducting bipartite lattices

Georges Bouzerar$^\star$ and Maxime Thumin

Université Grenoble Alpes, CNRS, Institut NEEL, F-38042 Grenoble, France

$\star$ georges.bouzerar@neel.cnrs.fr

## Abstract

Unconventional flat band (FB) superconductivity, as observed in van der Waals heterostructures, could open promising avenues towards high-$T_c$ materials. In FBs, pairings and superfluid weight scale linearly with the interaction parameter, such an unusual behaviour justifies and encourages strategies to promote FB engineering. Bipartite lattices (BLs) which naturally host FBs could be particularly interesting candidates. Within Bogoliubov de Gennes theory and in the framework of the attractive Hubbard model in BLs, a hidden symmetry of the quasi-particle eigenstates is revealed. As a consequence, we demonstrate universal relations for the pairings and the superfluid weight that are independent of the characteristics of the hopping term. Remarkably, it is shown that these general properties are insensitive to disorder as long as the bipartite character is protected.

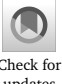
# 1 Introduction

Over the past decade, one can witness a growing interest for a novel family of emerging materials: The flat band (FB) systems [1–5]. In FB compounds, the kinetic energy being quenched, the electron-electron interaction energy becomes the unique relevant energy scale which gives an access to strongly correlated physics. FBs are found at the origin of a plethora of many body physics including an unconventional form of superconductivity (SC) of interband/geometric nature. In FBs, both the superfluid weight (SFW) and the critical temperature ($T_c$) scale linearly with the effective interaction amplitude $|U|$ [6–8] which contrasts with the dramatic $e^{-1/(\rho_F|U|)}$ scaling that characterizes the standard BCS theory, where $\rho_F$ is the density of states at the Fermi energy. Recently, it has been shown that the Bogoliubov de Gennes (BdG) approach is astonishingly quantitatively accurate in describing SC in FBs even in low-dimensional systems. Indeed, the SFW obtained with BdG formalism and that calculated with density matrix renormalization group (DMRG) were found to agree impressively in the sawtooth chain, the Creutz ladder and other quasi one-dimensional FB systems [9]. It is surprising to achieve such an agreement in the case of one-dimensional systems where quantum fluctuations are expected to be the strongest and hence have the most dramatic effects. Few years ago, considering the case of on-site attractive electron-electron interactions, it has been proven rigorously that the BCS wavefunction is an exact zero-temperature ground-state if FBs are isolated and under the condition that the pairings are uniform [8, 10, 11]. More recently, the validity of the BCS groundstate at zero temperature has been strengthened by a set of unbiased and numerically exact determinant quantum Monte Carlo (QMC) studies [12–15]. Indeed, in these articles, it has been unambiguously and nicely confirmed that both the SFW and $T_c$ scale perfectly linearly with the on-site Hubbard interaction parameter in the case of isolated FBs. In particular, in Ref. [14] it has been even shown that the SFW calculated within BdG approach agrees perfectly well with that obtained from QMC calculation. On the basis of such a series of crucial and numerically accurate results, both for one-dimensional and two-dimensional systems one can reasonably argue that despites its mean field nature the BdG approach is both qualitatively and quantitatively reliable for addressing superconductivity in two-dimensionnal FB systems.

In the framework of the on-site attractive Hubbard model (AHM) in bipartite lattices (BLs) and within BdG theory, our goal is to demonstrate universal sum-rules and other relations that pairings and SFW obey in half-filled and partially filled FB systems. We emphasize the fact that some of the relations proven here in a general context have been established for particular two-dimensional BLs assuming uniform pairings in each sublattice [8, 16]. For the sake of concreteness, all the properties that will be proven are illustrated in Appendix A,[1] with a typical asymmetric two-dimensional BL. Finally, in this work, we restrict ourself to $T = 0\,K$.

# 2 The BdG Hamiltonian

The bipartite lattice consists in two sublattices $\mathcal{A}$ and $\mathcal{B}$, which contain respectively $\Lambda_A$ and $\Lambda_B$ orbitals per cell. These two sets of orbitals are denoted $\mathcal{A} = \{A_1, A_2, \ldots, A_{\Lambda_A}\}$ and $\mathcal{B} = \{B_1, B_2, \ldots, B_{\Lambda_B}\}$. In the non-interacting case, the spectrum consists exactly in $N_{fb} = \Lambda_B - \Lambda_A$ flat bands located at $E = 0$ and $2\Lambda_A$ dispersives bands (DBs) which are symmetric because of chiral symmetry. We define $\Lambda = \Lambda_A + \Lambda_B$, the total number of orbitals per cell. Here, we assume $\Lambda_B > \Lambda_A$ which implies that FB eigenstates have a nonvanishing weight

---

[1]The Appendix A illustrates the sum-rules and other properties in a specific bipartite lattice.

on $\mathcal{B}$-orbitals only. The AHM reads,

$$\hat{H} = \sum_{Ii,Jj,\sigma} t^{IJ}_{A_i B_j} c^{\dagger}_{IA_i,\sigma} c_{JB_j,\sigma} - |U| \sum_{Il,\lambda=A,B} \hat{n}_{I\lambda_l,\uparrow} \hat{n}_{I\lambda_l,\downarrow} - \mu \hat{N} , \tag{1}$$

where $I, J$ are cell indices, $A_i$ (resp. $B_j$) labels the orbitals in $\mathcal{A}$ (resp. $\mathcal{B}$). Because of the bipartite nature of the lattice, the only non vanishing hoppings are of the form $t^{IJ}_{A_i B_j}$. The operator $c^{\dagger}_{I\lambda_l,\sigma}$ creates an electron of spin $\sigma$, in the orbital $\lambda_l$ of the $I$-th cell and $\hat{n}_{I\lambda_l,\sigma} = c^{\dagger}_{I\lambda_l,\sigma} c_{I\lambda_l,\sigma}$ where $\lambda = A, B$. $|U|$ is the strength of the on-site electron-electron interaction and finally $\mu$ denotes the chemical potential.

Within BdG theory, both pairings and local occupations are calculated self-consistently assuming a paramagnetic ground-state, $\langle \hat{n}_{\lambda_l,\uparrow} \rangle_0 = \langle \hat{n}_{\lambda_l,\downarrow} \rangle_0 = \frac{1}{2} n_{\lambda_l}$, where $\langle \ldots \rangle_0$ means thermal average. The BdG Hamiltonian reads,

$$\hat{H}_{BdG} = \sum_{\mathbf{k}} \begin{bmatrix} \hat{\mathbf{C}}^{\dagger}_{\mathbf{k}\uparrow} & \hat{\mathbf{C}}_{-\mathbf{k}\downarrow} \end{bmatrix} \begin{bmatrix} \hat{h}^{\uparrow}_{\mathbf{k}} & \hat{\Delta} \\ \hat{\Delta}^{\dagger} & -\hat{h}^{\downarrow*}_{-\mathbf{k}} \end{bmatrix} \begin{bmatrix} \hat{\mathbf{C}}_{\mathbf{k}\uparrow} \\ \hat{\mathbf{C}}^{\dagger}_{-\mathbf{k}\downarrow} \end{bmatrix} , \tag{2}$$

where the $\Lambda$-dimensional spinor $\hat{\mathbf{C}}^{\dagger}_{\mathbf{k}\sigma} = \left( \hat{\mathbf{C}}^{A\dagger}_{\mathbf{k}\sigma}, \hat{\mathbf{C}}^{B\dagger}_{\mathbf{k}\sigma} \right)^t$ with $\hat{\mathbf{C}}^{\lambda\dagger}_{\mathbf{k}\sigma} = (\hat{c}^{\dagger}_{\mathbf{k}\lambda_1,\sigma}, \hat{c}^{\dagger}_{\mathbf{k}\lambda_2,\sigma}, \ldots, \hat{c}^{\dagger}_{\mathbf{k}\lambda_{\Lambda_\lambda},\sigma})^t$ and $\lambda = A, B$. Finally, $\hat{c}^{\dagger}_{\mathbf{k}\lambda_l,\sigma}$ is the Fourier transform (FT) of $\hat{c}^{\dagger}_{I\lambda_l,\sigma}$ and,

$$\hat{h}^{\sigma}_{\mathbf{k}} = \begin{bmatrix} -\mu \hat{\mathbb{1}}_{\Lambda_A} - \hat{V}^A & \hat{h}_{AB} \\ \hat{h}^{\dagger}_{AB} & -\mu \hat{\mathbb{1}}_{\Lambda_B} - \hat{V}^B \end{bmatrix} . \tag{3}$$

$\hat{h}_{AB}$ is the FT of the tight-binding term in Eq. (1), the potential matrix is $\hat{V}^\lambda = \frac{|U|}{2} \text{diag}(n_{\lambda_1}, n_{\lambda_2}, \ldots, n_{\lambda_{\Lambda_\lambda}})$. Last, the pairing matrix is given by,

$$\hat{\Delta} = \begin{bmatrix} \hat{\Delta}^A & 0_{\Lambda_A \times \Lambda_B} \\ 0_{\Lambda_B \times \Lambda_A} & \hat{\Delta}^B \end{bmatrix} , \tag{4}$$

where $\hat{\Delta}^\lambda = \text{diag}(\Delta^\lambda_1, \Delta^\lambda_2, \ldots, \Delta^\lambda_{\Lambda_\lambda})$, and $\Delta^\lambda_l = -|U| \langle \hat{c}_{I\lambda_l,\downarrow} \hat{c}_{I\lambda_l,\uparrow} \rangle_0$. At half filling which corresponds to $\mu = -|U|/2$, the density being uniform [17] the diagonal blocks in Eq.(3) vanish. In addition, we assume time reversal symmetry which implies that the pairings can be taken real. From now on, for any finite $|U|$, the pairings are real and positive.

## 3 Hidden symmetry in the BdG eigenstates

The aim of the present section is to bring to light key properties of the BdG quasi-particle eigenstates. We emphasize that to the best of our knowledge the symmetry of the QP eigenstates reaveled in this section has never been mentioned in the literature till now. Before we proceed, we first need to define some necessary notations and definitions. Let us call positive (respectively negative) eigenstates those with positive (respectively negative) energy. Consider $|\Psi\rangle = (|\mathbf{u}\rangle, |\mathbf{v}\rangle)^t$ a BdG eigenstate of energy $E$, where $|\mathbf{u}\rangle = (|\mathbf{a}\rangle, |\mathbf{b}\rangle)^t$ and $|\mathbf{v}\rangle = (|\bar{\mathbf{a}}\rangle, |\bar{\mathbf{b}}\rangle)^t$. $|\mathbf{a}\rangle$ and $|\bar{\mathbf{a}}\rangle$ (respectively $|\mathbf{b}\rangle$ and $|\bar{\mathbf{b}}\rangle$) are column of length $\Lambda_A$ (respectively $\Lambda_B$).

We now propose to show that positive (respectively negative) eigenstates can be divided into two subsets $\mathcal{S}_+$ and $\mathcal{S}_-$, where, $|\Psi\rangle \in \mathcal{S}_+ \Leftrightarrow |\mathbf{v}\rangle = (|\mathbf{a}\rangle, -|\mathbf{b}\rangle)^t$, and $|\Psi\rangle \in \mathcal{S}_- \Leftrightarrow |\mathbf{v}\rangle = (-|\mathbf{a}\rangle, |\mathbf{b}\rangle)^t$.

At half-filling $\hat{H}_{BdG}$ is invariant under particle-hole (PH) transformation which reads,

$$\begin{bmatrix} \hat{\mathbf{C}}^{\dagger}_{A\uparrow} \\ \hat{\mathbf{C}}^{\dagger}_{B\downarrow} \end{bmatrix} \overset{PH}{\Longrightarrow} \begin{bmatrix} \hat{\mathbf{C}}_{A\downarrow} \\ -\hat{\mathbf{C}}_{B\uparrow} \end{bmatrix} . \tag{5}$$

As a consequence, the QP eigenstate $|\Psi\rangle = (|\mathbf{a}\rangle, |\mathbf{b}\rangle, |\bar{\mathbf{a}}\rangle, |\bar{\mathbf{b}}\rangle)^t$ becomes after PH transformation $|\Psi_1\rangle = (|\bar{\mathbf{a}}\rangle, -|\bar{\mathbf{b}}\rangle, |\mathbf{a}\rangle, -|\mathbf{b}\rangle)^t$. The PH symmetry implies that $|\Psi_1\rangle = e^{i\varphi}|\Psi\rangle$, which leads to $e^{i\varphi} = \pm 1$. Thus, we are left with two possibilities: (1) $|\Psi\rangle \in \mathcal{S}_+$ or (2) $|\Psi\rangle \in \mathcal{S}_-$ corresponding respectively to $\varphi = 0$ and $\varphi = \pi$. It is interesting to remark that, if $|\Psi_+\rangle$ of energy $E$ denotes an eigenstate in $\mathcal{S}_+$, then $\hat{U}|\Psi_+\rangle$ belongs to $\mathcal{S}_-$ and has energy $-E$, since $\hat{U}\hat{H}_{BdG}\hat{U}^\dagger = -\hat{H}_{BdG}$ where the unitary matrix $\hat{U} = \begin{bmatrix} 0 & \hat{\mathbb{1}}_\Lambda \\ -\hat{\mathbb{1}}_\Lambda & 0 \end{bmatrix}$.

We propose to proceed further and demonstrate a second interesting property of the BdG eigenstates that is crucial for what follows. For any finite $|U|$, the subset $\mathcal{S}_-$ (respectively $\mathcal{S}_+$) consists *exactly* in $\Lambda_B$ (respectively $\Lambda_A$) eigenstates of positive or zero energy and $\Lambda_A$ (respectively $\Lambda_B$) eigenstates of strictly negative energy.

In order to demonstrate this property, we first define for any given square hermitian matrix $\hat{M}$, $In(\hat{M}) = (n_m, n_p)$ where $n_m$ is the number of strictly negative eigenvalues and $n_p$ that of the positive or zero eigenvalues. Now, consider $|\phi_n^s\rangle = (|u_n^s\rangle, |v_n^s\rangle)^t$ a QP eigenstate of energy $E_n^s$ which belongs to $\mathcal{S}_s$ ($s = \pm$). Using Eq.(2) one can write,

$$\hat{\mathcal{H}}^s |u_n^s\rangle = E_n^s |u_n^s\rangle, \tag{6}$$

where the $\Lambda \times \Lambda$ matrices are,

$$\hat{\mathcal{H}}^+ = \begin{bmatrix} \hat{\Delta}^A & \hat{h}_{AB} \\ \hat{h}_{AB}^\dagger & -\hat{\Delta}^B \end{bmatrix}, \qquad \hat{\mathcal{H}}^- = \begin{bmatrix} -\hat{\Delta}^A & \hat{h}_{AB} \\ \hat{h}_{AB}^\dagger & \hat{\Delta}^B \end{bmatrix}. \tag{7}$$

For infinitesimal $|U|$, let us apply a degenerate perturbation theory to the set of $N_{fb}$ flat band eigenstates of $\hat{\mathcal{H}}^\pm|_{|U|=0}$ which, we recall have a finite weight on $\mathcal{B}$ orbitals only. The projection of $\hat{\Delta}^B$ in the FB eigenspace being positive definite, it implies (i) that the energy shift of each FB eigenstates of $\hat{\mathcal{H}}^+$ is strictly negative, (ii) while it is strictly positive for those of $\hat{\mathcal{H}}^-$. In other words, this means that $In(\hat{\mathcal{H}}^-) = (\Lambda_A, \Lambda_B)$ and $In(\hat{\mathcal{H}}^+) = (\Lambda_B, \Lambda_A)$.

To end the proof, assume that there exists a peculiar value $|U_c|$ such that for any $|U| < |U_c|$, $In(\hat{\mathcal{H}}^-) = (\Lambda_A, \Lambda_B)$ (resp. $In(\hat{\mathcal{H}}^+) = (\Lambda_B, \Lambda_A)$), while for $|U| > |U_c|$, $In(\hat{\mathcal{H}}^-) = (\Lambda_A + 1, \Lambda_B - 1)$ (resp. $In(\hat{\mathcal{H}}^+) = (\Lambda_B - 1, \Lambda_A + 1)$). At $|U_c|$, this would necessarily imply that both $\hat{\mathcal{H}}^-$ and $\hat{\mathcal{H}}^+$ have at least one zero energy eigenstate which we denote $|u_0^s\rangle = (|\mathbf{a}_0^s\rangle, |\mathbf{b}_0^s\rangle)^t$, where $s = \pm$. Consider the case $s = +$, from Eq.(7) we can write,

$$\begin{aligned} (\hat{\Delta}^B + \hat{h}_{AB}^\dagger (\hat{\Delta}^A)^{-1} \hat{h}_{AB})|\mathbf{b}_0^+\rangle &= 0, \\ |\mathbf{a}_0^+\rangle &= -(\hat{\Delta}^A)^{-1} \hat{h}_{AB} |\mathbf{b}_0^+\rangle. \end{aligned} \tag{8}$$

The fact that $\Delta_i^A > 0$ for any $i$ has been used. The matrix $\hat{\Delta}^B + \hat{h}_{AB}^\dagger (\hat{\Delta}^A)^{-1} \hat{h}_{AB}$ is the sum of a positive definite one and positive semi-definite one, hence their sum is positive definite. Thus, zero cannot be an eigenvalue implying that $|\mathbf{b}_0^+\rangle = [\mathbf{0}]_{\Lambda_B}$ and $|\mathbf{a}_0^+\rangle = [\mathbf{0}]_{\Lambda_A}$ where $|\mathbf{0}]_N$ is the column vector with $N$ zeros. The same procedure applies to $|u_0^-\rangle$ and completes the proof of this second interesting symmetry of the QP eigenstates.

# 4 Pairing sum rule in half-filled bipartite lattices

In this section, as a consequence of the QP eigenstates symmetry, our purpose is to demontrate general sum rules that pairings obey in any half-filled BL. In addition, we establish several bounds for the average pairings.

Focus first on the negative eigenstates of $\hat{H}_{BdG}$ and define $|\psi_{n+}^<\rangle$ where $n = 1, \ldots, \Lambda_B$ the normalized ones in $\mathcal{S}_+$ and similarly $|\psi_{m-}^<\rangle$ where $m = 1, \ldots, \Lambda_A$ those in $\mathcal{S}_-$. At $T = 0$,

pairings are given by,

$$\Delta_l^\lambda = -\frac{|U|}{N_c}\Big(\sum_{\mathbf{k},s=n+}\langle\psi_s^<|\hat{O}_{\lambda_l}|\psi_s^<\rangle + \sum_{\mathbf{k},s=m-}\langle\psi_s^<|\hat{O}_{\lambda_l}|\psi_s^<\rangle\Big), \tag{9}$$

where $\hat{O}_{\lambda_l} = \hat{c}_{-\mathbf{k}\lambda_l,\downarrow}\hat{c}_{\mathbf{k}\lambda_l,\uparrow}$, $\lambda = A, B$ and $l = 1,\ldots,\Lambda_\lambda$. The index $n$ runs over $1,\ldots,\Lambda_B$, and $m$ over $1,\ldots,\Lambda_A$, finally $N_c$ is the number of cells. Eq.(9) can be re-written,

$$\begin{aligned}\Delta_i^A &= -\frac{|U|}{N_c}\left(\sum_{\mathbf{k},n=1}^{n=\Lambda_B}|a_{ni}^+|^2 - \sum_{\mathbf{k},m=1}^{m=\Lambda_A}|a_{mi}^-|^2\right),\\ \Delta_j^B &= \frac{|U|}{N_c}\left(\sum_{\mathbf{k},n=1}^{n=\Lambda_B}|b_{nj}^+|^2 - \sum_{\mathbf{k},m=1}^{m=\Lambda_A}|b_{mj}^-|^2\right),\end{aligned} \tag{10}$$

where the fact that $|\psi_{n+}^<\rangle = (|a_n^+\rangle, b_n^+\rangle, |a_n^+\rangle, -|b_n^+\rangle)^t$ and $|\psi_{m-}^<\rangle = (|a_m^-\rangle, b_m^-\rangle, -|a_m^-\rangle, |b_m^-\rangle)^t$ has been used. These two sets of eigenstates being normalized, one finally ends up with the following sum-rule,

$$\sum_{j=1}^{\Lambda_B}\Delta_j^B - \sum_{i=1}^{\Lambda_A}\Delta_i^A = \frac{|U|}{2}(\Lambda_B - \Lambda_A). \tag{11}$$

A similar expression has been obtained recently in Ref. [18] where the uniform pairing condition has been assumed, and reads $\Delta_i^A = \Delta_A$ for any orbital $A_i$ in $\mathcal{A}$ and $\Delta_j^B = \Delta_B$ for any $B_j$ in $\mathcal{B}$. This hypothesis allows great simplifications in the calculations but does not correspond in general (presence of inequivalent orbitals) to the true self-consistent BdG solution. We underline the fact that in our general proof, the crucial step is the introduction of a hidden symmetry which allows us to split the BdG eigenstates into two subsets $\mathcal{S}_+$ and $\mathcal{S}_-$. We now propose to show some additional properties that arise from these results. Using Eq.(10), for any $B_j$ in $\mathcal{B}$, one gets $\Delta_j^B \leq \frac{|U|}{N_c}(\sum_{\mathbf{k},n=1}^{n=\Lambda_B}|b_{nj}^+|^2 + \sum_{\mathbf{k},m=1}^{m=\Lambda_A}|b_{mj}^-|^2) = |U|\langle\hat{n}_{B_j,\uparrow}\rangle = \frac{|U|}{2}$. Similarly, for any $i$, one finds $\Delta_i^A \leq \frac{|U|}{2}$.

If $\langle\Delta^\lambda\rangle$, $\lambda = A, B$, denotes the average of the pairings on each sublattice, then,

$$|U|(\langle\Delta^B\rangle - \langle\Delta^A\rangle) = \frac{1}{\Lambda_B}(F_1 - F_2), \tag{12}$$

where respectively $F_1 = \frac{1}{N_c}\sum_{\mathbf{k},j,n}|b_{nj}^+|^2 + \frac{r}{N_c}\sum_{\mathbf{k},i,n}|a_{ni}^+|^2$ and $F_2 = \frac{1}{N_c}\sum_{\mathbf{k},j,m}|b_{mj}^-|^2 + \frac{r}{N_c}\sum_{\mathbf{k},i,m}|a_{mi}^-|^2$, where $r = \frac{\Lambda_B}{\Lambda_A} \geq 1$ is introduced. Eigenstates being normalized, implies (i) $F_1 \geq \frac{\Lambda_B}{2}$ and (ii) $F_2 \leq \frac{\Lambda_B}{2}$ which finally leads to the inequality,

$$\langle\Delta_B\rangle \geq \langle\Delta_A\rangle. \tag{13}$$

In addition, combining this equation and Eq.(11) gives,

$$\frac{\langle\Delta_B\rangle}{|U|} \geq \frac{r-1}{2r}. \tag{14}$$

As an example, we consider the case of the stub lattice ($r = 2$). Recently, it has been found numerically that the lower bound of $\frac{\langle\Delta_B\rangle}{|U|}$ is 0.25 which coincides exactly with $\frac{r-1}{2r}$ [19]. Additionally, the sum-rule as given in Eq.(11) and the resulting inequalities are illustrated in Appendix A[1] in the case of a two-dimensional BL.

# 5 Pairings and occupations in partially filled flat bands

In the previous section we have discussed the case of half-filled FBs, we propose now to derive the pairings and local occupations in the case where these bands are partially filled which corresponds to an electron density $\nu$ that varies between $\nu_{min} = 2\Lambda_A$ and $\nu_{max} = 2\Lambda_B$.

To derive these expressions we rely on the pseudo-spin SU(2) symmetry of the AHM in BLs [20–22], which can interpreted as a form of rotation invariance in particle-hole space. Indeed, the AHM can be re-written,

$$\hat{H} = \sum_{Ii,Jj,\sigma} t_{A_iB_j}^{IJ} \hat{c}_{IA_i,\sigma}^{\dagger} \hat{c}_{JB_j,\sigma} - \frac{2}{3}|U| \sum_{I,l\lambda=A,B} \hat{\mathbf{T}}_{I\lambda_l} \cdot \hat{\mathbf{T}}_{I\lambda_l} - (\mu + |U|/2) \sum_{I,l,\lambda=A,B} \hat{n}_{I\lambda_l}. \tag{15}$$

The components of the local pseudo-spin operators read,

$$\hat{T}_{I\lambda_l}^{+} = \eta_\lambda \hat{c}_{I\lambda_l,\uparrow} \hat{c}_{I\lambda_l,\downarrow}, \tag{16}$$

$$\hat{T}_{I\lambda_l}^{-} = \eta_\lambda \hat{c}_{I\lambda_l,\downarrow}^{\dagger} \hat{c}_{I\lambda_l,\uparrow}^{\dagger}, \tag{17}$$

$$\hat{T}_{I\lambda_l}^{z} = \frac{1}{2}(1 - \hat{n}_{I\lambda_l}), \tag{18}$$

where $\eta_\lambda = 1$ (respectively $-1$) if $\lambda = A$ (respectively $B$). These operators obey the usual commutation relations of spin operators. Notice that in partially filled FBs, the last term (right side) in Eq.(15) vanishes and implies that $[\widehat{H}, \hat{T}^{\pm}] = [\widehat{H}, \hat{T}^z] = 0$, where $\hat{\mathbf{T}} = \sum_{I,l,\lambda=A,B} \hat{\mathbf{T}}_{I\lambda_l}$ is the total pseudo-spin operator. Thus, for $\mu = -|U|/2$ the Hamiltonian has a pseudospin SU(2) symmetry.

The average of the local pseudo-spin operator $\langle \hat{\mathbf{T}}_{I\lambda_l} \rangle_0$ is cell independent and reads,

$$\langle \hat{\mathbf{T}}_{\lambda_l} \rangle_0 = \begin{bmatrix} \langle \hat{T}_{\lambda_l}^x \rangle_0 = \eta_\lambda \Re(\frac{\Delta_l^\lambda}{|U|}) \\ \langle \hat{T}_{\lambda_l}^y \rangle_0 = \eta_\lambda \Im(\frac{\Delta_l^\lambda}{|U|}) \\ \langle \hat{T}_{\lambda_l}^z \rangle_0 = \frac{1}{2}(1 - n_{\lambda_l}) \end{bmatrix}. \tag{19}$$

$\hat{H}_{BdG}$ is invariant under any identical rotation of the pseudo-spins. It is convenient to consider $\mathcal{R}_y(\theta)$ the rotation of angle $\theta$ around the $y$-axis. This transformation results in,

$$\begin{bmatrix} \hat{c}_{I\lambda_l,\uparrow} \\ \hat{c}_{I\lambda_l,\downarrow} \end{bmatrix} \overset{\mathcal{R}_y(\theta)}{\Longrightarrow} \begin{bmatrix} \cos(\theta/2)\hat{c}_{I\lambda_l,\uparrow} - \eta_\lambda \sin(\theta/2)\hat{c}_{I\lambda_l,\downarrow}^{\dagger} \\ \cos(\theta/2)\hat{c}_{I\lambda_l,\downarrow} + \eta_\lambda \sin(\theta/2)\hat{c}_{I\lambda_l,\uparrow}^{\dagger} \end{bmatrix}. \tag{20}$$

Consider that the self-consistent solution for the half-filled case $\nu = \bar{\nu} = \Lambda_A + \Lambda_B$ is known. The expectation value of the corresponding pseudo-spins reads,

$$\bar{\mathbf{T}}_{\lambda_l} = \begin{bmatrix} \bar{T}_{\lambda_l}^x = \eta_\lambda \frac{\bar{\Delta}_l^\lambda}{|U|} \\ \bar{T}_{\lambda_l}^y = 0 \\ \bar{T}_{\lambda_l}^z = 0 \end{bmatrix}. \tag{21}$$

Both $\bar{T}_{\lambda_l}^y$ and $\bar{T}_{\lambda_l}^z$ vanish since (i) the pairings are taken real and (ii) because of the uniform density theorem [17]. The application of $\mathcal{R}_y(\theta)$ to the pseudo-spins leads to a BdG solution corresponding to a partial filling of the FBs where,

$$\Delta_l^\lambda = \bar{\Delta}_l^\lambda \cos(\theta), \tag{22}$$

$$n_{\lambda_l} = 1 + 2\eta_\lambda \frac{\bar{\Delta}_l^\lambda}{|U|} \sin(\theta). \tag{23}$$

For a given $\theta$, the corresponding filling factor is,

$$\nu(\theta) = \overline{\nu} + \sin(\theta)(\Lambda_B - \Lambda_A). \tag{24}$$

It should be stressed that the sum-rule of Eq.(11) has been used. Hence, $\theta = \pi/2$ corresponds to the fully filled FBs, i.e. $\nu = \nu_{max}$ while $\theta = -\pi/2$ to empty FBs or $\nu = \nu_{min}$. Combining Eq.(22) and Eq.(24), one obtains the local occupations and pairings for any partial filling of the FBs,

$$\Delta_l^\lambda = \bar{\Delta}_l^\lambda f(\nu), \tag{25}$$

$$n_\lambda = 1 \pm 2\eta_\lambda \frac{\bar{\Delta}_l^\lambda}{|U|} \sqrt{1 - f^2(\nu)}, \tag{26}$$

where $+$ (respectively $-$) corresponds to $\nu \geqslant \bar{\nu}$ (respectively $\nu \leqslant \bar{\nu}$), and

$$f(\nu) = \frac{2}{\nu_{max} - \nu_{min}} \sqrt{(\nu - \nu_{min})(\nu_{max} - \nu)}. \tag{27}$$

Similar expressions have been derived in Ref. [18], where a uniform pairing is forced on the orbitals on the dominant lattice. Our proof is general, without restriction on the pairings, and requires only that the sum-rule given in Eq.(11) has been proven.

# 6 The superfluid weight in partially filled FBs

The superfluid weight being the key quantity that characterizes the superconducting phase, we propose in this section to derive a general relationship between the SFW in partially filled FBs and that of half-filled BL. The SFW along the $\mu$-direction is defined as [23, 24],

$$D_\mu^s = \frac{1}{N_c} \frac{\partial^2 \Omega(\mathbf{q})}{\partial q_\mu^2}\Big|_{\mathbf{q}=\mathbf{0}}, \tag{28}$$

where $\Omega(\mathbf{q})$ is the grand-potential and the momentum $\mathbf{q}$ mimics the effect of a vector potential introduced by a standard Peierls substitution.

Recently, it has been argued that when the quantum metric (QM) [25, 26] associated to FBs is not minimal, corrections should be included in Eq .(28) [18]. Contrary to $D_\mu^s$, the QM which measures the typical spreading of the FB eigenstates is a quantity which depends on the orbital positions. However, for any BL, one can always find the orbital positions inside the cell that minimize the QM (optimal positions), therefore, for which Eq.(28) is correct. In general, it corresponds to the most symmetrical positions of the orbitals in the cell.

Assuming a minimal QM and following Refs. [8] and [16] the SFW can be written,

$$D_\mu^s = \frac{2}{N_c} \sum_{\mathbf{k},mn} \frac{J_\mu^{nm}}{E_n^< - E_m^>}, \tag{29}$$

where $J_\mu^{nm} = |\langle \Psi_n^< | \hat{V}_\mu | \Psi_m^> \rangle|^2 - |\langle \Psi_n^< | \hat{\Gamma} \hat{V}_\mu | \Psi_m^> \rangle|^2$, with $\hat{\Gamma} = \text{diag}(\hat{\mathbb{1}}_{\Lambda \times \Lambda}, -\hat{\mathbb{1}}_{\Lambda \times \Lambda})$ and $\hat{V} = \text{diag}(\hat{v}^0, \hat{v}^0)$.

The velocity operator along the $\mu$-direction is $\hat{v}_\mu^0 = \frac{\partial \hat{h}^0}{\partial k_\mu}$ where $\hat{h}^0 = \begin{bmatrix} 0 & \hat{h}_{AB} \\ \hat{h}_{AB}^\dagger & 0 \end{bmatrix}$.

In addition, to avoid any confusion due to multiple indices, we introduce the notation $|\Psi_m^>\rangle = (|a_m^>\rangle, |b_m^>\rangle, |\bar{a}_m^>\rangle, |\bar{b}_m^>\rangle)^t$ for the positive eigenstates, and $|\Psi_n^<\rangle = (|a_n^<\rangle, |b_n^<\rangle, |\bar{a}_n^<\rangle, |\bar{b}_n^<\rangle)^t$ for the negative ones. Their respective energy are $E_m^>$ and $E_n^<$. For the moment, we can ignore

whether these states belong to $\mathcal{S}^{\pm}$. The eigenstates for $\nu = \bar{\nu}$ are specified by simply replacing $n \to 0n$ and $m \to 0m$.

Assuming the QP eigenstates known for $\nu = \bar{\nu}$, the superfluid weight $D_\mu^s$ in partially filled FBs is obtained using the pseudospin SU(2) symmetry of the Hamiltonian discussed previously. We recall that the eigenvalues of the QP states are invariant under this transformation. From Eq.(20) the rotated eigenstates are, $|\psi_n^<\rangle = \hat{U}_\theta|\Psi_{0n}^<\rangle$ (similarly $|\Psi_m^>\rangle = \hat{U}_\theta|\Psi_{0m}^>\rangle$) where the unitary transformation reads,

$$\hat{U}_\theta = \begin{bmatrix} c & 0 & s & 0 \\ 0 & c & 0 & -s \\ -s & 0 & c & 0 \\ 0 & s & 0 & c \end{bmatrix}, \tag{30}$$

where $c = \cos(\theta/2)$ and $s = \sin(\theta/2)$.

The calculation of the matrix elements $J_\mu^{nm}$ in Eq.(29) require that of,

$$\langle \Psi_n^<|\hat{\Gamma}^p\hat{V}_\mu|\Psi_m^>\rangle = \langle \Psi_{0n}^<|\hat{U}_{-\theta}\hat{\Gamma}^p\hat{V}_\mu\hat{U}_\theta|\Psi_{0m}^>\rangle, \tag{31}$$

where $p = 0$ or 1.

According to the BdG symmetry of the eigenstates one can write, $|\bar{a}_{0n}^<\rangle = \epsilon_n|a_{0n}^<\rangle$ and $|\bar{b}_{0n}^<\rangle = -\epsilon_n|b_{0n}^<\rangle$ where $\epsilon_n = 1$ (respectively $-1$) if $|\Psi_{0n}^<\rangle \in \mathcal{S}^+$ (respectively $\in \mathcal{S}^-$). We proceed similarly with the positive eigenstates $|\Psi_{0m}^>\rangle$ and get,

$$J_\mu^{nm} = |C_{0nm}^{<,>}|^2 g_{nm}, \tag{32}$$

where $g_{nm} = ((1-\epsilon_n\epsilon_m)c+(\epsilon_n+\epsilon_m)s)^2-(1+\epsilon_n\epsilon_m)^2$ and $C_{0nm}^{<,>} = \langle a_{0n}^<|\partial_\mu\hat{h}_{AB}|b_{0m}^>\rangle+\langle b_{0n}^<|\partial_\mu\hat{h}_{AB}^\dagger|a_{0m}^>\rangle$. Eq.(32) can be simplified and leads to,

$$J_\mu^{nm} = -4\epsilon_n\epsilon_m|C_{0nm}^{<,>}|^2\cos^2(\theta). \tag{33}$$

Using Eq.(24), we finally end up with,

$$D_\mu^s(\nu) = f^2(\nu)D_\mu^s(\bar{\nu}). \tag{34}$$

In partially filled FBs, $D_\mu^s$ always has a universal parabolic shape and vanishes for $\nu = \nu_{min}$ and $\nu_{max}$. We remark that the derivation of Eq.(34) in a general context, requires the sum-rule given in Eq.(11). It is as well interesting to note that Eq.(33) indicates that the contributions to $D_\mu^s(\nu)$ originating from pairs of eigenstates in the same subspace $\mathcal{S}^+$ or $\mathcal{S}^-$ are positive, while they are negative in the other case.

# 7 Effects of disorder

We have previously considered the case of clean systems. An interesting question is: What is the impact of disorder that preserves the bipartite character of the lattice such as random hoppings or introduction of vacancies? Translation invariance being broken, $\hat{H}_{BdG}$ must be diagonalized in real space. The number of zero energy eigenstates is $\mathcal{N}_{E=0} = |\mathcal{N}_\mathcal{B} - \mathcal{N}_\mathcal{A}|$ where $\mathcal{N}_\lambda$ is the total number of orbitals in each sublattice ($\lambda = \mathcal{A}, \mathcal{B}$). In the clean case, our proofs are based on the hidden symmetry of the BdG eigentates which remain valid in the single cell made up of $\mathcal{N}_\mathcal{A}$ A-orbitals and $\mathcal{N}_\mathcal{B}$ B-orbitals. As a consequence, in the disordered half-filled BL, Eq.(11) simply becomes,

$$\sum_{j=1}^{\mathcal{N}_\mathcal{B}}\Delta_j^B - \sum_{i=1}^{\mathcal{N}_\mathcal{A}}\Delta_i^A = \frac{|U|}{2}|\mathcal{N}_\mathcal{B} - \mathcal{N}_\mathcal{A}|, \tag{35}$$

where $i$ (respectively $j$) now runs over the whole sublattice $\mathcal{A}$ (respectively $\mathcal{B}$). In addition, Eq.(25) and Eq.(34) which give the filling dependence of the pairings and the SFW remain valid as well.

Notice, as mentioned in the introduction, that BCS wavefunction being the exact ground-state in BL hosting isolated FBs when $|U|$ is smaller than the gap [8, 10, 11], should imply as well the exactness of our results in this limit. Thus, it would be of great interest to confirm this expectation from numerically exact methods such as DMRG, a reliable and well suited tool for quasi one-dimensional systems but as well from Quantum Monte Carlo studies which would be appropriate to address the case of two-dimensional BL lattices. It should be emphasized that our results are restricted to the case of on-site intractive interaction. It has been revealed in a recent QMC study that the effect of including a nearest neighbour attractive electron-electron interaction destabilize the superconducting phase and favors phase separation [12]. In the present work the focus was made on half-filled and partially filled FBs, however it worth mentioning that another interesting sum-rule emerges when the Fermi level lies in the dispersive bands. Indeed, it has been found numerically, in the case of the $\mathcal{L}-$ lattice, that for any position of the Fermi energy inside the DBs and for any $|U|$ the pairings satisfy $\Lambda_B \langle \Delta^B \rangle = \Lambda_A \langle \Delta^A \rangle$. This sum-rule has already been highlighted in a previous work devoted to FB superconductivity in the stub lattice [27] and seems to occur systematically in BLs.[2]

To conclude, using a hidden symmetry of the BdG eigenstates, we have rigorously demonstrated that in bipartite lattices the pairings and the SFW obey universal relations. Furthermore, these general properties are shown to hold in disordered systems as long as the bipartite character of the lattice is conserved. Our findings could have an important impact in the search of novel families of compounds exhibiting unconventional FB superconductivity.

# A  Supplemental material

In this supplemental material our purpose is to illustrate the sum-rules and other relations demonstrated in the general context of bipartite lattices (BLs) where flat bands (FBs) are either half-filled or partially filled. The prototype of two-dimensional BL considered here will be designated by $\mathcal{L}$-lattice. It is depicted in Fig.1(a). The $\mathcal{L}$-lattice consists in two sublattices $\mathcal{A}$ and $\mathcal{B}$ which contain respectively $\Lambda_A = 3$ and $\Lambda_B = 5$ orbitals per unit cell, where $\mathcal{A} = \{A_1, A_2, A_3\}$ and $\mathcal{B} = \{B_1, B_2, \dots, B_5\}$ . In the absence of electron-electron interaction, the one-particle spectrum consists exactly in $N_{fb} = \Lambda_B - \Lambda_A = 2$ FBs with energy $E_{fb} = 0$ and $2\Lambda_A = 6$ symmetric dispersives bands that result from chirality. The compact localized eigenstates (CLS) are depicted in Fig.1(b). As expected the weight is finite for the orbitals of sublattice $\mathcal{B}$ only. From these CLS one can easily build the corresponding Bloch FB eigenstates that are respectively,

$$|\Psi_1^{fb}\rangle = \begin{pmatrix} 0 \\ 0 \\ 0 \\ -b_{12} \\ b_{12} \\ b_{34} \\ -b_{34} \\ 0 \end{pmatrix}, \qquad |\Psi_2^{fb}\rangle = \begin{pmatrix} 0 \\ 0 \\ 0 \\ \beta_1 \\ \beta_2 \\ \beta_3 \\ \beta_4 \\ \beta_5 \end{pmatrix}, \tag{A.1}$$

---

[2]A rigourous proof within BdG theory is currently under investigation [28].

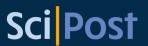

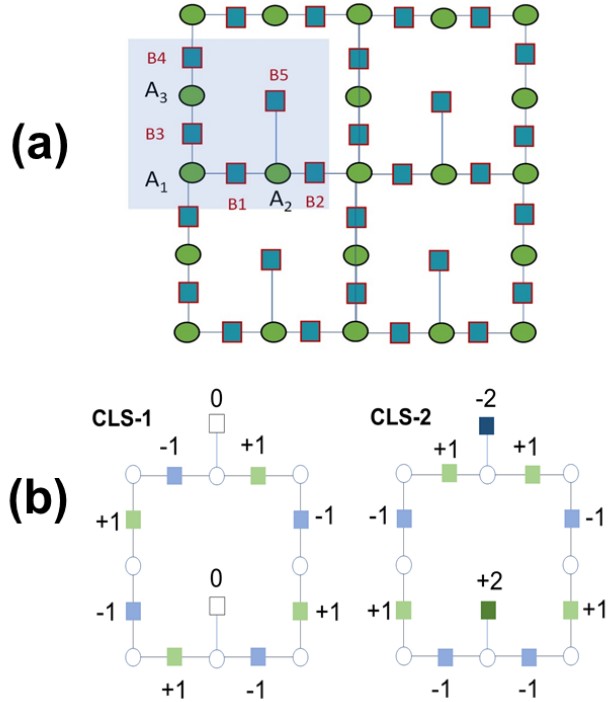

Figure 1: (a) Prototype of two-dimensional asymmetric bipartite lattice ($\mathcal{L}$), with $\Lambda_A = 3$ atoms of type A and $\Lambda_B = 5$ atoms of type B per unit cell depicted by the shaded area. The hoppings are restricted to nearest neighbors only, they are all equal and set to 1. The single particle Hamiltonian has two degenerate flat bands. (b) The compact localized FB eigensates of the $\mathcal{L}$-lattice. The amplitude is non vanishing for the orbitals of the sublattice $\mathcal{B}$ only.

where,

$$
\begin{aligned}
b_{12} &= e^{ik_y/2} \sin(k_y/2), \\
b_{34} &= e^{ik_x/2} \sin(k_x/2),
\end{aligned}
\tag{A.2}
$$

and,

$$
\begin{aligned}
\beta_1 &= +e^{ik_y/2} \sin(k_y/2)(\sin(k_x) + i f_{xy}), \\
\beta_2 &= -e^{ik_y/2} \sin(k_y/2)(\sin(k_x) - i f_{xy}), \\
\beta_3 &= +2e^{ik_x/2} \sin^2(k_y/2) \cos(k_x/2), \\
\beta_4 &= -2e^{ik_x/2} \sin^2(k_y/2) \cos(k_x/2), \\
\beta_5 &= -2i e^{ik_y/2} \sin(k_y/2) f_{xy},
\end{aligned}
\tag{A.3}
$$

where $f_{xy} = 2(\sin^2(k_x/2) + \sin^2(k_y/2))$. Notice that these two eigenstates are not normalized. These expressions indicate unambiguously that the pairings associated with the orbitals of the sublattice $\mathcal{B}$ should be non-uniform. However, as illustrated in the next paragraph they are identical for $(B_1, B_2)$ and $(B_3, B_4)$ pairs which could be anticipated from the lattice symmetry.

## A.1 Symmetry in the $H_{BdG}$ eigenstates

It has been shown in our manuscript, that eigenstates of the Bogoliubov de Gennes Hamiltonian $H_{BdG}$ as defined in Eq.(2) in the main text, can be divided into two subsets $\mathcal{S}_+$ and $\mathcal{S}_-$ which are defined in what follows.
Consider a normalized eigenstate of $H_{BdG}$, denoted $|\Psi\rangle = (|u\rangle, |v\rangle)^t$, where $|u\rangle = (|\mathbf{a}\rangle, |\mathbf{b}\rangle)^t$

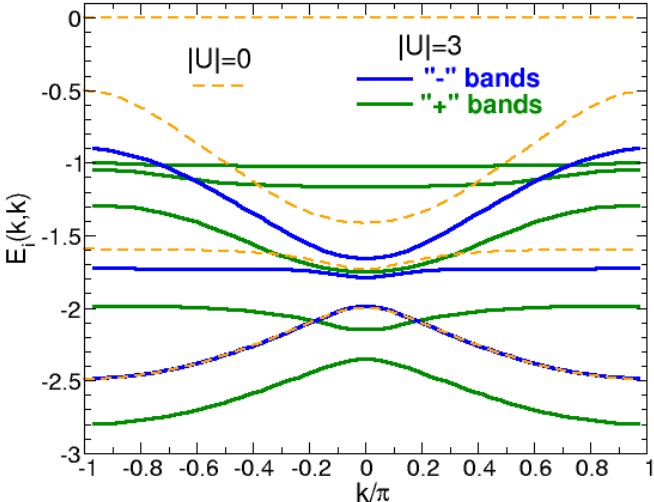

Figure 2: Negative part of the quasiparticle dispersions along the $(1,1)$-direction for the half-filled $\mathcal{L}$ lattice as depicted in Fig.1. The green (respectively blue) continuous lines correspond to QP eigenstates in $\mathcal{S}_+$ (respectively $\mathcal{S}_-$). As expected there are $\Lambda_A = 3$ bands in $\mathcal{S}_-$ and $\Lambda_B = 5$ in $\mathcal{S}_+$. Here, the on-site interaction parameter $|U| = 3$, the conclusion remains the same for any $|U|$. The (orange) dashed lines represent the dispersions in the case of the non interacting system ($|U| = 0$), each bands being doubly degenerate.

and $|\nu\rangle = (|\bar{\mathbf{a}}\rangle, |\bar{\mathbf{b}}\rangle)^t$, the column vectors $|\mathbf{a}\rangle$ and $|\bar{\mathbf{a}}\rangle$ (respectively $|\mathbf{b}\rangle$ and $|\bar{\mathbf{b}}\rangle$) being of length $\Lambda_A$ (respectively $\Lambda_B$),

$$|\Psi\rangle \in \mathcal{S}_+ \Leftrightarrow |\bar{\mathbf{a}}\rangle = |\mathbf{a}\rangle\,, \qquad |\bar{\mathbf{b}}\rangle = -|\mathbf{b}\rangle\,, \tag{A.4}$$

$$|\Psi\rangle \in \mathcal{S}_- \Leftrightarrow |\bar{\mathbf{a}}\rangle = -|\mathbf{a}\rangle\,, \quad |\bar{\mathbf{b}}\rangle = |\mathbf{b}\rangle\,. \tag{A.5}$$

For a given value of the on-site electron-electron interaction strength $|U|$, here $|U| = 3$ has been chosen, Fig.2 depicts the QP dispersions in the half-filled $\mathcal{L}-$lattice and along the $\Gamma M$ direction in the Brillouin zone. We show only the negative part of spectrum. Unambiguously, for any value of the momentum $\mathbf{k}$, the spectrum consists exactly in $\Lambda_A = 3$ eigenstates which belong to $\mathcal{S}_-$ and $\Lambda_B = 5$ eigenstates to $\mathcal{S}_+$.

### A.2 Sum rule for the pairings in half-filled bipartite lattices

In the main text of the article, it has been rigorously proven in our BdG theory that in half-filled bipartite lattices the pairings obey the following sum-rule,

$$\sum_{j=1}^{\Lambda_B} \Delta_j^B - \sum_{i=1}^{\Lambda_A} \Delta_i^A = \frac{|U|}{2}(\Lambda_B - \Lambda_A)\,. \tag{A.6}$$

In Fig.3, the local pairings for each orbital are plotted as a function of $|U|$ in the case of the half-filled $\mathcal{L}$-lattice. For obvious symmetry reasons (see Fig.1), one finds that $\Delta_1^B = \Delta_2^B$ and $\Delta_3^B = \Delta_4^B$ while the A-type pairings are all different. As it can be clearly seen, for any $|U|$, Eq.(A.6) is exactly fulfilled.

If we now define the average value of the pairing on both sublattices by $\langle \Delta_\lambda \rangle$ where $\lambda = A, B$. For any $|U|$, we have shown in the main text that,

$$\langle \Delta_B \rangle \geq \langle \Delta_A \rangle\,, \tag{A.7}$$

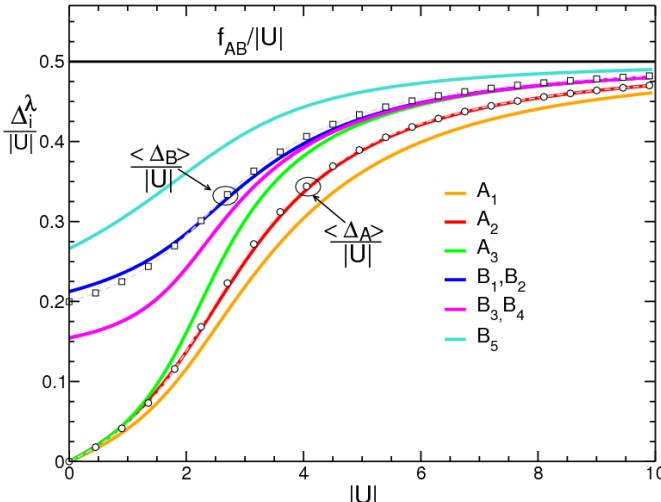

Figure 3: Rescaled pairings as a function of $|U|$ in the half-filled BL depicted in Fig.1. The open circles (respectively squares) are the average values of the pairings on sublattice $\mathcal{A}$ (respectively $\mathcal{B}$). The horizontal black line corresponds to $f_{AB}/|U|$ where

$$f_{AB} = \frac{1}{2}\left(\sum_{j=1}^{\Lambda_B} \Delta_j^B - \sum_{i=1}^{\Lambda_A} \Delta_i^A\right).$$

and found as well a lower bound for the average value of the pairings on $\mathcal{B}$-sublattice,

$$\frac{\langle \Delta_B \rangle}{|U|} \geq \frac{r-1}{2r}, \tag{A.8}$$

where $r = \Lambda_B/\Lambda_A$ has been introduced.

First, for any $|U|$, Fig.3 clearly shows that $\langle \Delta_B \rangle \geq \langle \Delta_A \rangle$. In addition, according to Eq.(A.8), one expects that $\langle \Delta_B \rangle \geq 0.2\,|U|$ which is in perfect agreement with the results depicted in Fig.3. More precisely, the lower bound is found to coincide exactly with $\left.\frac{\partial \langle \Delta_B \rangle}{\partial |U|}\right|_{U=0}$.

## A.3 The superfluid weight in partially filled FBs

In the main text we have proved a general relationship between the superfluid weight (SFW) $D_\mu^s$ in partially filled FBs and that of the half-filled lattice. The SFW is defined as [23, 24],

$$D_\mu^s = \frac{1}{N_c} \left.\frac{\partial^2 \Omega(\mathbf{q})}{\partial q_\mu^2}\right|_{\mathbf{q}=\mathbf{0}}, \tag{A.9}$$

where $\Omega(\mathbf{q})$ is the grand-potential and $\mathbf{q}$ mimics the effect of a vector potential, introduced by a standard Peierls substitution in the hopping terms in the BdG Hamiltonian.

First, we should emphasize the fact that we have carefully checked in the case of the $\mathcal{L}$-lattice that (i) the corrections to Eq.(A.9) as discussed in Ref. [18] are vanishing and (ii) the quantum metric associated to the FBs is minimal for the geometry depicted in Fig.1.
Using the pseudo-spin SU(2) symmetry of the Hamiltonian for $\mu = -|U|/2$ [20–22], it has been shown in the main text that,

$$D_\mu^s(\nu) = f^2(\nu) D_\mu^s(\bar{\nu}), \tag{A.10}$$

where the filling dependent function is,

$$f(\nu) = \frac{2}{\nu_{max} - \nu_{min}} \sqrt{(\nu - \nu_{min})(\nu_{max} - \nu)}. \tag{A.11}$$

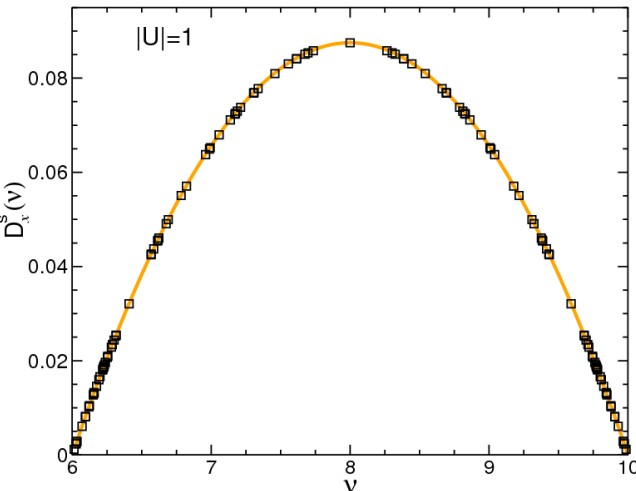

Figure 4: Superfluid weight $D_x^s$ as a function of the electron filling in the $\mathcal{L}$-lattice. The densities correspond to partially filled FBs where the chemical potential is $\mu = -|U|/2$. The electron interaction parameter is $|U| = 1$. The symbols are the numerical data and the continuous line is the analytical calculation as discussed in the main text and given in Eq.(34).

Thus, the SFW for partially filled FBs always has a universal parabolic shape and $D_\mu^s(\nu)$ vanishes for $\nu = \nu_{min}$ and $\nu = \nu_{max}$. These fillings correspond respectively to empty FBs for which $\nu = \nu_{min} = 2\Lambda_A = 6$ and fully filled FBs where $\nu = \nu_{max} = 2\Lambda_B = 10$. Fig.4 depicts the SFW $D_x^s(\nu)$ as a function of the electron density $\nu$ in the $\mathcal{L}$-lattice. As it is clearly seen, the agreement between the numerical data and the analytical expression given in Eq.(A.10) is excellent.

### A.4 The impact of the disorder: The case of randomly distributed vacancies

In our manuscript, it has been argued that in the presence of disorder that conserves the bipartite character of the lattice the sum-rules and other relations established in the case of clean systems still hold. Here, our goal is to illustrate this feature. We consider the impact of vacancies randomly distributed in the $\mathcal{L}-$lattice. In the main text it was predicted that in a half-filled disordered system, Eq.(11) would become,

$$\sum_{j=1}^{\mathcal{N}_\mathcal{B}} \Delta_j^B - \sum_{i=1}^{\mathcal{N}_\mathcal{A}} \Delta_i^A = \frac{|U|}{2} |\mathcal{N}_\mathcal{B} - \mathcal{N}_\mathcal{A}|, \tag{A.12}$$

where $i$ (respectively $j$) runs now over the whole sublattice $\mathcal{A}$ (respectively $\mathcal{B}$), and $\mathcal{N}_\mathcal{A}$ (resp. $\mathcal{N}_\mathcal{B}$) are the total number of A-orbitals (respectively B-orbitals) in the disordered lattice.

Because of the loss of translation invariance, the calculations require now multiple real space diagonalizations of the BdG Hamiltonian, until convergence in the self-consistent loop is reached. The size of the matrices is $2\mathcal{N} \times 2\mathcal{N}$ where $\mathcal{N} = \mathcal{N}_\mathcal{A} + \mathcal{N}_\mathcal{B}$. For our illustration, we have chosen a sufficiently large system that contains 3200 orbitals. In Fig.5, the pairing distributions in the disordered half-filled $\mathcal{L}$-lattice are plotted. The configuration of disorder corresponds to the introduction of 5% of vacancies randomly distributed. We have checked that Eq.(A.12) is exactly verified (high accuracy), as well as the relation $\langle \Delta_B \rangle \geq \langle \Delta_A \rangle$ which could be already anticipated from the plot of the pairing distributions. Additionally, we have

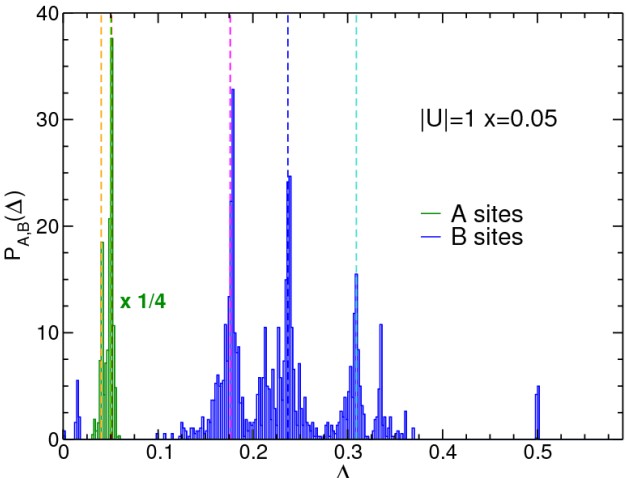

Figure 5: Pairing distributions $P_\Lambda(\Delta)$ for both sublattices ($\Lambda = A, B$) in the disordered $\mathcal{L}$-lattice. The concentration of randomly distributed vacancies $x = 0.05$, the pristine system contains $\mathcal{N} = 8 \times 20^2$ orbitals. The system is half-filled and $|U| = 1$. For more visibility the probability distribution $P_A(\Delta)$ has been multiplied by $1/4$. The vertical dashed lines are the pairings in the case of the clean lattice.

checked that Eq.(A.8) is as well fulfilled $\frac{\langle \Delta_B \rangle}{|U|} \geq \frac{1}{2}(1 - \frac{1}{r})$, where in the disordered lattice $r = \frac{\mathcal{N}_\mathcal{B}}{\mathcal{N}_\mathcal{A}}$.

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
