# Peer review of "Hidden symmetry of Bogoliubov de Gennes quasi-particle eigenstates and universal relations in flat band superconducting bipartite lattices"

_SciPost Physics, doi:SciPost Phys. Core 7, 018 (2024)_

## Round 2 · Referee Report · Anonymous · 2024-2-27

Report
The authors improved their manuscript and better contextualized it in the existing literature.
Their results are valid within Bogoliubov-de Gennes theory for bipartite lattices at zero temperature. Lieb theorem and existing numerical studies suggest that the mean-filed description might be appropriate for an isolated flat band with a strictly local attractive Hubbard interaction. A result similar to Eq.11 was obtained in the literature under the assumption of uniform pairing. The main contribution of the manuscript is the extension of this formula to bipartite lattices that do not satisfy the uniform pairing conditions. The authors now correctly stress all these points and accurately present the contribution of their work.
This submission does not meet the acceptance criteria upheld by SciPost Physics. In light of the limited applicability of its results, it does not present a breakthrough in a long-standing research block or a groundbreaking discovery. Instead, it meets the criteria of SciPost Physics Core, where it could be published. Before acceptance, I hope the authors can discuss in the supplemental material the single-particle physics of the lattice model considered. In particular, they should discuss the Wannier states of the two flat bands to ensure that they have weight on all B sublattices, for example, and violate the uniform pairing condition.

---

## Round 2 · Referee Report · Anonymous · 2024-3-5

Strengths
Shows explicit computations and demonstrates results for a class of concrete problems
Weaknesses
In my mind the applicability to materials (claimed in the final sentence for example) remains somewhat limited.
Minor: I think there are few typos e.g. "interactive interaction" for attractive interaction.
Report
The authors have revised the paper in light of comments from the first round of referee reports, and the manuscript has thus improved. However, I do not believe that it reaches the level appropriate to SciPost Physics since it does not either "Present a breakthrough on a previously-identified and long-standing research stumbling block", or "
Open a new pathway in an existing or a new research direction, with clear potential for multipronged follow-up work". However, it does provide an original approach to an old problem and advances current understanding. As such I believe it can be published in SciPost Physics Core.

---

## Round 2 · Author Response

List of changes
Dear Editor,
find below our response to the reviewer's criticisms and comments.
Sincerely yours,
* * *
Report 1
-Referee:This manuscript studies some aspects of superconductivity, specifically various "universal" relations between the pairing amplitudes and superfluid stiffness, in “flat-band” systems in the presence of *only* an on-site attractive-U Hubbard interaction, and assuming (without any justification; see below) that mean-field theory is applicable. Unfortunately, the paper does make some misleading statements (perhaps unintentionally), especially when it comes to presenting results using the said mean-field theory as if they hold well beyond this uncontrolled and unjustified limit.
Response: It is incorrect and unfair to say that no justification that mean-field theory is applicable has been provided. Indeed, as was explicitly written highlighted in the previous version of our manuscript (in the introduction), an excellent agreement was found between the BdG approach and the numerically exact and unbiased DMRG in various one-dimensional systems for both the calculations of the pairings and that of the superfluid weight. In addition, as it was emphasized, the agreements found were extremely impressive. Because in one dimensional systems quantum fluctuations are the strongest, the agreement revealed in these studies is a clear signature that the BdG approach captures not only qualitatively but quantitatively as well the physics (when the Fermi level lies in a FB).
Hence, it is natural to expect its reliability as well in 2D systems. We have now extended the introduction to discuss several recent QMC studies performed in 2D systems which further support the validity of our mean field approach.
In addition, we stress that we did not make any misleading statement either intentionally or unintentionally.
We have improved, modified in depth the previous version of our manuscript in order to insist further on the fact that our calculations are obtained in the framework of the BdG approach (mean field) and to discuss the connection of our present study with recent QMC studies.
-Referee: At the same time, the paper fails to cite many important papers, including an early paper by Tovmasyan et al, Phys. Rev. B 94, 245149 (2016), which beautifully highlighted many important aspects of the flat-band problem with on-site attractive-U Hubbard interaction, including where BCS theory is valid. As shown by these authors, the only limit where the BCS wavefunction is the actual ground-state wavefunction, thereby providing some a posteriori justification, is tied to an emergent SU(2) symmetry in the problem. Furthermore, they also derived relations between the superfluid stiffness and pairing amplitudes (they are both related to the interaction strength, as well as some other invariants).
Response: This reference is now cited in the new version of our manuscript with several other works concerning QMC studies on superconductivity in flat bands.
The relations derived in PRB 94, 245149 between pairings and SFW concern the case of uniform pairings as already discussed in the previous version of the manuscript (where a different reference was quoted). In addition, in PRB 94, 245149 it is shown that the BCS wave function is the exact ground state if a uniform pairing condition on the single-particle states is satisfied, and that the compressibility is diverging as a consequence of an emergent SU(2) symmetry. However, this does not mean that when pairings are not uniform the validity of the BdG approach breaks down. Similarly, we cannot say that the pairing sum rule proven in early studies under the condition of uniform pairings is valid in this case only. The DMRG studies strongly and unambiguously support the fact that the mean field approach is reliable and accurate even in the case of 1D systems (strong quantum fluctuations) and even when the pairings are not uniform, which is the case in the sawtooth chain.
-Referee: I do not find this manuscript well written/presented and contributing any fundamental new insights into the problem (though it tries to present results in a “rigorous” fashion, even when the starting point of the approach lacks much rigor). Ultimately, all of the calculations involve various manipulations of the same BdG equations. At this point, based on the large existing literature of similar analysis of the FB problem using mean-field theory, I find that this work is just a straightforward extension of these ideas.
Response: It is unfair and incorrect to pretend that there is nothing new in our work. The crucial hidden symmetry of the BdG eigenstates revealed here has never been shown in any existing published papers in the literature. If the referee has a relevant reference where this has been done then we should definitely quote it because it is one of the key features in our work. Here, we are able to derive relations between pairings and SFW in a very general way without the necessity to force a uniform pairing on the same sublattice. Even in the presence of a small amount of disorder in a system such as the Lieb lattice this condition breaks down. In addition, we are able to demonstrate that the proven relations hold even in the presence of disorder which, to the best of our knowledge, has never been done as well. Our article has been extensively modified in order to better underline its approach, its results and link them with existing literature, with existing QMC studies in particular.
-Referee: Interestingly, there is repeated mention of how well BCS mean-field theory apparently agrees with DMRG. However, as has been pointed out in various unbiased, large-scale two-dimensional QMC simulations, BCS mean-field theory also fails in dramatic ways once the on-site attraction is perturbed by infinitesimal nearest-neighbor interactions (Hofmann et al., ‘20, ‘23; Peri et al., ‘21). The authors do not cite any of these papers as well.
Response: We have now quoted several QMC studies including these references and have discussed how these unbiased numerically exact studies support the BdG calculations performed in our work. We would like to mention that Referee 3 has suggested as well to quote these references in order, I quote her/him, “ to better argue for the validity of our mean field approach”.
Regarding the effect of including a next nearest neighbor electron-electron interaction (attractive in the case of Hofmann et al.,), such a study goes beyond the scope of the present manuscript. However, we have quoted this point in the novel version of our manuscript.
* * *
Report 2
-Referee: The key step is the identification of a "hidden symmetry" in the BdG eigenstates, i.e. a property of the mean-field solutions
Response: Indeed, the hidden symmetry revealed in our manuscript is one of the key features. To underline this point and avoid any confusion, the title of our manuscript has been modified.
-Referee: I am not persuaded of the validity of the results beyond the mean-field limit assumed without any clear justification Indeed, there are convincing results (see e.g. PRB 94, 245149) suggesting that the exact BCS wavefunction and hence the BdG solutions are only valid in some special limits.
Response: In this reference (now quoted in the new version of our manuscript) it is shown that the BCS wave function is the exact ground state of the projected interaction Hamiltonian, if (1) a uniform pairing condition on the single-particle states is satisfied, and (2) if the single particle FBs are well separated from the dispersive bands. To the best of our knowledge, there is no proof that the BCS wavefunction is the exact ground state if these two conditions are not fulfilled. We believe that the condition (2) is important. However, recent studies indicate that the uniform pairing condition (1) may not be essential. Indeed, in non bipartite lattices such as the Creutz ladder, where pairings are not uniform, unbiased DMRG calculations have revealed an excellent agreement with the BdG calculations for both the pairings (see Fig.7 in Ref. PRB 105, 024502) and the superfluid weight as well (Fig.4 in Ref. PRB 105, 024502). Furthermore the agreement found is excellent for any value of |U| (from weak to strong coupling regime).
Thus, It would be of great interest in the near future to compare our findings with those that could be obtained from numerically exact methods such as QMC or DMRG. Remark that proofs given in our manuscript are not restricted to 2D systems, hence the sum-rules could be tested easily in quasi one dimensional systems.
-Referee: Furthermore given that bipartite lattices often admit sign-free Quantum Monte Carlo there exist unbiased numerical studies of the attractive-U Hubbard problem on such lattices even with flat bands (e.g. PRB 102, 201112), but no effort is made by the present authors to compare to these results or address their implications for their work. This lack of any embedding of their work into the very intense and broader range of activity in the field to my mind actually raises the issue to one of poor writing, and as such it is very difficult to assess the significance or originality of the work (which is not to immediately say that it is unoriginal or insignificant, only that a reader is forced to suspend their judgement in the absence of evidence.)
Response: We agree that references to QMC studies were probably missing in the previous version of our manuscript. Hence, efforts have been made in the novel version of our article to correct this issue (especially in the introduction) We now cite and discuss several recently published Quantum Monte Carlo studies that concern the unconventional superconductivity in Flat bands. Indeed, we have added several references including that quoted by the referee: PRB 102 201112, PRL 130 226001, PRL 128 087002, and PRL 126 027002.
-Referee: At minimum, the authors must explain the extent to which their results are valid outside the mean-field model, or else reframe the paper as a series of exact results on the mean field model, in order to be able to make a fair assessment on whether the paper may be suitable I believe there may exist some argument to be made that new exact results on a mean field model could be of interest but the bar is high, and if there is not a plausible case to be made of either the importance of these new results or for their validity beyond mean field my recommendation would be to reject; however I am willing to consider that authors may be able to rebut these criticisms in a major revision.
Response: We had already specified several times in the previous version that the results obtained fell within the framework of the BdG approach, so we have decided to insist more by highlighting this fact in the modified title and in the abstract. Our calculations are obtained within BdG theory which is indeed a mean field approach. However, as it was said, the agreement between BdG and the numerically exact DMRG for both pairings and superfluid weight is impressive (PRB 105, 024502) in the case of 1D systems (even not bipartite lattices). This is clearly a strong indication that quantum fluctuations are already properly treated within the mean field approach. Furthermore, in the Lieb lattice, it has been shown in PRL 117, 045303 that BdG theory and DMFT (respectively ED) agree quantitatively at low temperature for both the calculations of the pairings and of the Superfluid weight. In addition, the comparison with unbiased QMC studies in various 2D systems has clearly shown that the SFW and the critical temperature scales linearly with the |U| in agreement again with BdG theory. As an additional exemple, in PRL128 087002 (reference quoted by Referee3) it has been shown that the QMC calculations of the ratio Ds/|U| (|U| being smaller than the single particle gap) at T=0K agree perfectly with the BdG result as nicely illustrated in Fig.3 of this letter.
Therefore, on the basis of these remarkable results, it is definitely reasonable to think, to argue that the results shown in our study could be valid beyond the mean-field. It will be necessary to consider QMC or DMRG studies to confirm it. We hope that our work will motivate such studies. It might be straightforward to show the sum rules in simple quasi-1D bipartite lattices. An improved discussion is now available in the new version of our manuscript.
* * *
Report 3
-Referee: Their findings agree with previous results reported in the literature and are not entirely novel. The main advancement is the identification of a symmetry in the BdG equations on bipartite lattices. This symmetry allows them to extend previous results beyond the uniform pairing condition, i.e., the assumption of equal pairing on all sublattices on which the flat band eigenstates have non-zero weight.
Response: We have properly quoted in the previous version of the manuscript the studies where the uniform pairing condition is assumed. This is now further emphasized in the novel version of our paper. Yes, the key feature here is the identification of a hidden symmetry never mentioned in the existing literature that allows to demonstrate general relations even in the absence of translation invariance (presence of disorder).
-Referee: What is lacking and necessary is a better framing in the existing literature. In particular, the authors should better address the regime of validity of their findings. PRB 94, 245149, and Nat. Comm. 6, 8944 show how the BCS wavefunction is an exact zero-temperature ground state of isolated flat bands with on-site attractive interactions. While the latter is restricted to bipartite lattices, the first extends this result beyond bipartite lattices with uniform pairing. Further evidence for the validity of the BCS approximation at zero temperature is provided by the quantum Monte Carlo (QMC) results of PRB 102, 201112; PRL 130, 226001; PRL 128, 087002; PRL 126, 027002. The authors could use the above references to better argue for the validity of their mean-field approach.
Response: We agree with the reviewer that references to QMC studies were missing. We have now improved our manuscript in order to mention this relevant series of QMC studies. We agree as well with the fact that these sets of numerically unbiased studies allow to reinforce the validity of the BdG approach used in our present study.
Referee: The authors should address at least the following questions:
a-What happens without time-reversal symmetry? Which result will break down?
Response: This is indeed an interesting question. In the absence of TRS, we cannot anymore assume real positive pairings which is a crucial element for the proofs given in the present work. In the absence of TRS, the pairings would be complex numbers where the phases cannot be removed by any gauge transformation. So far, it is unclear to us what will happen in this case but it is very likely that the sum-rules for the modulus of the pairings will break down. This is an interesting open problem that could be investigated in the near future by considering for instance the impact of a magnetic field.
b-What will happen in the presence of intra-sublattice pairing or longer-range interactions?
Response: We guess that the referee meant “presence of intra-sublattice hopping”, which will break the bipartite character of the lattice, and hence lead to quasi flat bands (dispersive). We expect the relations for the pairings and SFW to break down. To get quantitative numbers would require a complete numerical study. The symmetry obtained for the QP eigenstates will break down.
But, if the referee meant “inter-sublattice pairing” then it is related to the effects of extended electron-electron interaction. The presence of longer-range interaction such as nearest neighbor couplings goes beyond the scope of the present study. To perform such a study within BdG, one would have to include in the decoupling scheme in addition to the on-site cooper pairs, pairs on nearest neighbors sites. There is no doubt that this would certainly be an interesting study to carry out. However, the inclusion of a nearest neighbor attractive interaction term has been addressed within the QMC approach in Hofmann et al. 2020 (this reference is now quoted), where it has been shown that the presence of such a term destabilizes the superconducting phase and favors phase separation.
c-They should distinct between gapless and gapped flat bands. In the latter, the mean-field treatment seems better justified.
Response: We do not have to specify whether the system is gapless or gapped to prove the hidden symmetry of the QP eigenstates. Hence, it has no impact on the relations derived.
d- Which results are strictly valid for the flat bands and which for any band of the
bipartite lattice?
Response: We thank the referee for this relevant and interesting question. The relations proven in our manuscript, we mean those between the pairings and the superfluid weight, concern exclusively the case of flat bands. In fact, we expect in the case where the Fermi level lies in the dispersive bands a different relation. As it has been pointed out recently in the case of the one-dimensional stub lattice (EPL 144 56001). One expects for dispersive bands the following relation: <ΔB>/<ΔA> = NA/NB. This has been numerically checked in the case of the 2D lattice studied in the present manuscript and in other two dimensional lattices. The rigorous proof of this novel relation is currently under investigation. We have decided to add a few sentences regarding this point in the novel version of our manuscript.
e-Lastly, to meet the criteria upheld by sciPost, the authors should specify the relevance of their findings. I think they could stress more the importance of extending previous results beyond the uniform pairing condition.
Response: We thought that this was clear enough in the previous version of our paper. However, we have followed the referee's suggestion and stressed it further in the new version of the manuscript.
Referee: Some minor remarks:
- The authors should present the non-interacting band structure of the tight-binding model studied in the supplemental material.
Response: We have modified Fig.2. The quasi-particle dispersions for the non-interacting system are now plotted on the same figure.
-There seems to be an issue with the citation of equations and references. For example, below Eq. (35) the authors refer to Eq. (7) of the supplemental material rather than Eq. (34) of the main text.
Similarly, below Eq. (11) they cite Ref. (1) rather than Ref. 13. Ref. 9 is repeated in Ref. 10
Response: We apologize and are sorry for this issue which was related to the presence in the same tex file of the supplementary material. These errors are now corrected in the novel version.

---

## Round 2 · List of Changes

Dear Editor,
find below our response to the reviewer's criticisms and comments.
Sincerely yours,
* * *
Report 1
-Referee:This manuscript studies some aspects of superconductivity, specifically various "universal" relations between the pairing amplitudes and superfluid stiffness, in “flat-band” systems in the presence of *only* an on-site attractive-U Hubbard interaction, and assuming (without any justification; see below) that mean-field theory is applicable. Unfortunately, the paper does make some misleading statements (perhaps unintentionally), especially when it comes to presenting results using the said mean-field theory as if they hold well beyond this uncontrolled and unjustified limit.
Response: It is incorrect and unfair to say that no justification that mean-field theory is applicable has been provided. Indeed, as was explicitly written highlighted in the previous version of our manuscript (in the introduction), an excellent agreement was found between the BdG approach and the numerically exact and unbiased DMRG in various one-dimensional systems for both the calculations of the pairings and that of the superfluid weight. In addition, as it was emphasized, the agreements found were extremely impressive. Because in one dimensional systems quantum fluctuations are the strongest, the agreement revealed in these studies is a clear signature that the BdG approach captures not only qualitatively but quantitatively as well the physics (when the Fermi level lies in a FB).
Hence, it is natural to expect its reliability as well in 2D systems. We have now extended the introduction to discuss several recent QMC studies performed in 2D systems which further support the validity of our mean field approach.
In addition, we stress that we did not make any misleading statement either intentionally or unintentionally.
We have improved, modified in depth the previous version of our manuscript in order to insist further on the fact that our calculations are obtained in the framework of the BdG approach (mean field) and to discuss the connection of our present study with recent QMC studies.
-Referee: At the same time, the paper fails to cite many important papers, including an early paper by Tovmasyan et al, Phys. Rev. B 94, 245149 (2016), which beautifully highlighted many important aspects of the flat-band problem with on-site attractive-U Hubbard interaction, including where BCS theory is valid. As shown by these authors, the only limit where the BCS wavefunction is the actual ground-state wavefunction, thereby providing some a posteriori justification, is tied to an emergent SU(2) symmetry in the problem. Furthermore, they also derived relations between the superfluid stiffness and pairing amplitudes (they are both related to the interaction strength, as well as some other invariants).
Response: This reference is now cited in the new version of our manuscript with several other works concerning QMC studies on superconductivity in flat bands.
The relations derived in PRB 94, 245149 between pairings and SFW concern the case of uniform pairings as already discussed in the previous version of the manuscript (where a different reference was quoted). In addition, in PRB 94, 245149 it is shown that the BCS wave function is the exact ground state if a uniform pairing condition on the single-particle states is satisfied, and that the compressibility is diverging as a consequence of an emergent SU(2) symmetry. However, this does not mean that when pairings are not uniform the validity of the BdG approach breaks down. Similarly, we cannot say that the pairing sum rule proven in early studies under the condition of uniform pairings is valid in this case only. The DMRG studies strongly and unambiguously support the fact that the mean field approach is reliable and accurate even in the case of 1D systems (strong quantum fluctuations) and even when the pairings are not uniform, which is the case in the sawtooth chain.
-Referee: I do not find this manuscript well written/presented and contributing any fundamental new insights into the problem (though it tries to present results in a “rigorous” fashion, even when the starting point of the approach lacks much rigor). Ultimately, all of the calculations involve various manipulations of the same BdG equations. At this point, based on the large existing literature of similar analysis of the FB problem using mean-field theory, I find that this work is just a straightforward extension of these ideas.
Response: It is unfair and incorrect to pretend that there is nothing new in our work. The crucial hidden symmetry of the BdG eigenstates revealed here has never been shown in any existing published papers in the literature. If the referee has a relevant reference where this has been done then we should definitely quote it because it is one of the key features in our work. Here, we are able to derive relations between pairings and SFW in a very general way without the necessity to force a uniform pairing on the same sublattice. Even in the presence of a small amount of disorder in a system such as the Lieb lattice this condition breaks down. In addition, we are able to demonstrate that the proven relations hold even in the presence of disorder which, to the best of our knowledge, has never been done as well. Our article has been extensively modified in order to better underline its approach, its results and link them with existing literature, with existing QMC studies in particular.
-Referee: Interestingly, there is repeated mention of how well BCS mean-field theory apparently agrees with DMRG. However, as has been pointed out in various unbiased, large-scale two-dimensional QMC simulations, BCS mean-field theory also fails in dramatic ways once the on-site attraction is perturbed by infinitesimal nearest-neighbor interactions (Hofmann et al., ‘20, ‘23; Peri et al., ‘21). The authors do not cite any of these papers as well.
Response: We have now quoted several QMC studies including these references and have discussed how these unbiased numerically exact studies support the BdG calculations performed in our work. We would like to mention that Referee 3 has suggested as well to quote these references in order, I quote her/him, “ to better argue for the validity of our mean field approach”.
Regarding the effect of including a next nearest neighbor electron-electron interaction (attractive in the case of Hofmann et al.,), such a study goes beyond the scope of the present manuscript. However, we have quoted this point in the novel version of our manuscript.
* * *
Report 2
-Referee: The key step is the identification of a "hidden symmetry" in the BdG eigenstates, i.e. a property of the mean-field solutions
Response: Indeed, the hidden symmetry revealed in our manuscript is one of the key features. To underline this point and avoid any confusion, the title of our manuscript has been modified.
-Referee: I am not persuaded of the validity of the results beyond the mean-field limit assumed without any clear justification Indeed, there are convincing results (see e.g. PRB 94, 245149) suggesting that the exact BCS wavefunction and hence the BdG solutions are only valid in some special limits.
Response: In this reference (now quoted in the new version of our manuscript) it is shown that the BCS wave function is the exact ground state of the projected interaction Hamiltonian, if (1) a uniform pairing condition on the single-particle states is satisfied, and (2) if the single particle FBs are well separated from the dispersive bands. To the best of our knowledge, there is no proof that the BCS wavefunction is the exact ground state if these two conditions are not fulfilled. We believe that the condition (2) is important. However, recent studies indicate that the uniform pairing condition (1) may not be essential. Indeed, in non bipartite lattices such as the Creutz ladder, where pairings are not uniform, unbiased DMRG calculations have revealed an excellent agreement with the BdG calculations for both the pairings (see Fig.7 in Ref. PRB 105, 024502) and the superfluid weight as well (Fig.4 in Ref. PRB 105, 024502). Furthermore the agreement found is excellent for any value of |U| (from weak to strong coupling regime).
Thus, It would be of great interest in the near future to compare our findings with those that could be obtained from numerically exact methods such as QMC or DMRG. Remark that proofs given in our manuscript are not restricted to 2D systems, hence the sum-rules could be tested easily in quasi one dimensional systems.
-Referee: Furthermore given that bipartite lattices often admit sign-free Quantum Monte Carlo there exist unbiased numerical studies of the attractive-U Hubbard problem on such lattices even with flat bands (e.g. PRB 102, 201112), but no effort is made by the present authors to compare to these results or address their implications for their work. This lack of any embedding of their work into the very intense and broader range of activity in the field to my mind actually raises the issue to one of poor writing, and as such it is very difficult to assess the significance or originality of the work (which is not to immediately say that it is unoriginal or insignificant, only that a reader is forced to suspend their judgement in the absence of evidence.)
Response: We agree that references to QMC studies were probably missing in the previous version of our manuscript. Hence, efforts have been made in the novel version of our article to correct this issue (especially in the introduction) We now cite and discuss several recently published Quantum Monte Carlo studies that concern the unconventional superconductivity in Flat bands. Indeed, we have added several references including that quoted by the referee: PRB 102 201112, PRL 130 226001, PRL 128 087002, and PRL 126 027002.
-Referee: At minimum, the authors must explain the extent to which their results are valid outside the mean-field model, or else reframe the paper as a series of exact results on the mean field model, in order to be able to make a fair assessment on whether the paper may be suitable I believe there may exist some argument to be made that new exact results on a mean field model could be of interest but the bar is high, and if there is not a plausible case to be made of either the importance of these new results or for their validity beyond mean field my recommendation would be to reject; however I am willing to consider that authors may be able to rebut these criticisms in a major revision.
Response: We had already specified several times in the previous version that the results obtained fell within the framework of the BdG approach, so we have decided to insist more by highlighting this fact in the modified title and in the abstract. Our calculations are obtained within BdG theory which is indeed a mean field approach. However, as it was said, the agreement between BdG and the numerically exact DMRG for both pairings and superfluid weight is impressive (PRB 105, 024502) in the case of 1D systems (even not bipartite lattices). This is clearly a strong indication that quantum fluctuations are already properly treated within the mean field approach. Furthermore, in the Lieb lattice, it has been shown in PRL 117, 045303 that BdG theory and DMFT (respectively ED) agree quantitatively at low temperature for both the calculations of the pairings and of the Superfluid weight. In addition, the comparison with unbiased QMC studies in various 2D systems has clearly shown that the SFW and the critical temperature scales linearly with the |U| in agreement again with BdG theory. As an additional exemple, in PRL128 087002 (reference quoted by Referee3) it has been shown that the QMC calculations of the ratio Ds/|U| (|U| being smaller than the single particle gap) at T=0K agree perfectly with the BdG result as nicely illustrated in Fig.3 of this letter.
Therefore, on the basis of these remarkable results, it is definitely reasonable to think, to argue that the results shown in our study could be valid beyond the mean-field. It will be necessary to consider QMC or DMRG studies to confirm it. We hope that our work will motivate such studies. It might be straightforward to show the sum rules in simple quasi-1D bipartite lattices. An improved discussion is now available in the new version of our manuscript.
* * *
Report 3
-Referee: Their findings agree with previous results reported in the literature and are not entirely novel. The main advancement is the identification of a symmetry in the BdG equations on bipartite lattices. This symmetry allows them to extend previous results beyond the uniform pairing condition, i.e., the assumption of equal pairing on all sublattices on which the flat band eigenstates have non-zero weight.
Response: We have properly quoted in the previous version of the manuscript the studies where the uniform pairing condition is assumed. This is now further emphasized in the novel version of our paper. Yes, the key feature here is the identification of a hidden symmetry never mentioned in the existing literature that allows to demonstrate general relations even in the absence of translation invariance (presence of disorder).
-Referee: What is lacking and necessary is a better framing in the existing literature. In particular, the authors should better address the regime of validity of their findings. PRB 94, 245149, and Nat. Comm. 6, 8944 show how the BCS wavefunction is an exact zero-temperature ground state of isolated flat bands with on-site attractive interactions. While the latter is restricted to bipartite lattices, the first extends this result beyond bipartite lattices with uniform pairing. Further evidence for the validity of the BCS approximation at zero temperature is provided by the quantum Monte Carlo (QMC) results of PRB 102, 201112; PRL 130, 226001; PRL 128, 087002; PRL 126, 027002. The authors could use the above references to better argue for the validity of their mean-field approach.
Response: We agree with the reviewer that references to QMC studies were missing. We have now improved our manuscript in order to mention this relevant series of QMC studies. We agree as well with the fact that these sets of numerically unbiased studies allow to reinforce the validity of the BdG approach used in our present study.
Referee: The authors should address at least the following questions:
a-What happens without time-reversal symmetry? Which result will break down?
Response: This is indeed an interesting question. In the absence of TRS, we cannot anymore assume real positive pairings which is a crucial element for the proofs given in the present work. In the absence of TRS, the pairings would be complex numbers where the phases cannot be removed by any gauge transformation. So far, it is unclear to us what will happen in this case but it is very likely that the sum-rules for the modulus of the pairings will break down. This is an interesting open problem that could be investigated in the near future by considering for instance the impact of a magnetic field.
b-What will happen in the presence of intra-sublattice pairing or longer-range interactions?
Response: We guess that the referee meant “presence of intra-sublattice hopping”, which will break the bipartite character of the lattice, and hence lead to quasi flat bands (dispersive). We expect the relations for the pairings and SFW to break down. To get quantitative numbers would require a complete numerical study. The symmetry obtained for the QP eigenstates will break down.
But, if the referee meant “inter-sublattice pairing” then it is related to the effects of extended electron-electron interaction. The presence of longer-range interaction such as nearest neighbor couplings goes beyond the scope of the present study. To perform such a study within BdG, one would have to include in the decoupling scheme in addition to the on-site cooper pairs, pairs on nearest neighbors sites. There is no doubt that this would certainly be an interesting study to carry out. However, the inclusion of a nearest neighbor attractive interaction term has been addressed within the QMC approach in Hofmann et al. 2020 (this reference is now quoted), where it has been shown that the presence of such a term destabilizes the superconducting phase and favors phase separation.
c-They should distinct between gapless and gapped flat bands. In the latter, the mean-field treatment seems better justified.
Response: We do not have to specify whether the system is gapless or gapped to prove the hidden symmetry of the QP eigenstates. Hence, it has no impact on the relations derived.
d- Which results are strictly valid for the flat bands and which for any band of the
bipartite lattice?
Response: We thank the referee for this relevant and interesting question. The relations proven in our manuscript, we mean those between the pairings and the superfluid weight, concern exclusively the case of flat bands. In fact, we expect in the case where the Fermi level lies in the dispersive bands a different relation. As it has been pointed out recently in the case of the one-dimensional stub lattice (EPL 144 56001). One expects for dispersive bands the following relation: <ΔB>/<ΔA> = NA/NB. This has been numerically checked in the case of the 2D lattice studied in the present manuscript and in other two dimensional lattices. The rigorous proof of this novel relation is currently under investigation. We have decided to add a few sentences regarding this point in the novel version of our manuscript.
e-Lastly, to meet the criteria upheld by sciPost, the authors should specify the relevance of their findings. I think they could stress more the importance of extending previous results beyond the uniform pairing condition.
Response: We thought that this was clear enough in the previous version of our paper. However, we have followed the referee's suggestion and stressed it further in the new version of the manuscript.
Referee: Some minor remarks:
- The authors should present the non-interacting band structure of the tight-binding model studied in the supplemental material.
Response: We have modified Fig.2. The quasi-particle dispersions for the non-interacting system are now plotted on the same figure.
-There seems to be an issue with the citation of equations and references. For example, below Eq. (35) the authors refer to Eq. (7) of the supplemental material rather than Eq. (34) of the main text.
Similarly, below Eq. (11) they cite Ref. (1) rather than Ref. 13. Ref. 9 is repeated in Ref. 10
Response: We apologize and are sorry for this issue which was related to the presence in the same tex file of the supplementary material. These errors are now corrected in the novel version.

---

## Round 3 · List of Changes

-The CLS eigenstates are now shown (in Fig.1)
-The analytic expression of the Flat band eigenstates of the one particle Hamiltonian( U=0)
are given.

---

## Editorial Decision

published